# Genome-Wide Transcriptome Analysis of Rice Seedlings after Seed Dressing with *Paenibacillus yonginensis* DCY84^T^ and Silicon

**DOI:** 10.3390/ijms20235883

**Published:** 2019-11-23

**Authors:** Yo-Han Yoo, Minjae Kim, Anil Kumar Nalini Chandran, Woo-Jong Hong, Hye Ryun Ahn, Gang Taik Lee, Sungju Kang, Dabin Suh, Jin-O Kim, Yeon-Ju Kim, Ki-Hong Jung

**Affiliations:** 1Graduate School of Biotechnology & Crop Biotech Institute, Kyung Hee University, Yongin 17104, Korea; directorhan@khu.ac.kr (Y.-H.Y.); tisia69@khu.ac.kr (M.K.); anikng@gmail.com (A.K.N.C.); hwj0602@khu.ac.kr (W.-J.H.); ahyeryun@naver.com (H.R.A.); gangtaik@hanmail.net (G.T.L.); 4supersonic@naver.com (S.K.); dabin304@naver.com (D.S.); kjo0659@naver.com (J.-O.K.); 2Graduate School of Biotechnology, College of Life Science, Kyung Hee University, Yongin 17104, Korea; yeonjukim@khu.ac.kr

**Keywords:** *Paenibacillus yonginensis* DCY84^T^, PGPB, RNA-Seq, Root, silicon, rice, drought stress

## Abstract

Plant-growth-promoting bacteria (PGPB) are beneficial microorganisms that can also protect against disease and environmental stress. Silicon (Si) is the second most abundant element in soil, and is known to increase plant growth, grain yield, resistance to biotic stress, and tolerance to abiotic stress. Combined treatment of PGPB and Si has been shown to further enhance plant growth and crop yield. To determine the global effects of the PGPB and Si on rice growth, we compared rice plants treated with *Paenibacillus yonginensis* DCY84^T^ (DCY84^T^) and Si with untreated rice. To identify the genes that respond to DCY84^T^+Si treatment in rice, we performed an RNA-Seq transcriptome analysis by sampling treated and untreated roots on a weekly basis for three weeks. Overall, 576 genes were upregulated, and 394 genes were downregulated in treated roots, using threshold fold-changes of at least 2 (log_2_) and *p*-values < 0.05. Gene ontology analysis showed that phenylpropanoids and the L-phenylalanine metabolic process were prominent in the upregulated genes. In a metabolic overview analysis using the MapMan toolkit, pathways involving phenylpropanoids and ethylene were strongly associated with upregulated genes. The functions of seven upregulated genes were identified as being associated with drought stress through a literature search, and a stress experiment confirmed that plants treated with DCY84^T^+Si exhibited greater drought tolerance than the untreated control plants. Furthermore, the predicted protein–protein interaction network analysis associated with DCY84^T^+ Si suggests mechanisms underlying growth promotion and stress tolerance.

## 1. Introduction

Modern intensive agriculture depends on synthetic chemical fertilizers containing essential plant nutrients such as nitrogen, phosphorus, and potassium [1,2]. However, excessive use of such fertilizers leads to the accumulation of insoluble phosphates in the soil, and also results in ecosystem disturbances and environmental pollution [3,4]. Recently, the inoculation of plant-growth-promoting bacteria (PGPB) into crops has been shown to improve crop growth and increase resistance from various environmental stresses [5]. In addition, PGPB have been reported to promote plant growth through mechanisms such as biological nitrogen fixation, phosphate solubilization, plant hormone regulation, and siderophore production [5,6,7]. PGPB, which are beneficial microorganisms found primarily on the root surface (“rhizobacteria”) that can also protect against disease and environmental stresses [8,9].

The genus *Paenibacillus* is a well-known member of the PGPB group, together with *Acetobacter*, *Azotobacter*, *Burkholderia*, and *Pseudomonas* [10]. To date, about 150 species of *Paenibacillus* have been identified (http://www.bacterio.net/paenibacillus.html); they are widespread, having been isolated from alkaline soil, ginseng field soil, rice field soil, and gamma-irradiated Antarctic soil [11,12,13,14]. *Paenibacillus yonginensis* DCY84^T^ (DCY84^T^) was found in the humus soil of Yongin forest in Gyeonggi-do, South Korea. It is a Gram-positive, rod-shaped, aerobic, spore-forming bacterium that is motile by means of peritrichous flagella [15]. In addition, DCY84^T^ has been reported to not only promote plant growth, but also to protect plants from biotic and abiotic stresses. For example, *Arabidopsis* treated with DCY84^T^ reportedly exhibited increased tolerance to salinity, drought, and heavy metal stresses [16].

Silicon (Si), the second most abundant element in soil [17], is classified as a “quasi-essential element for plant growth”, and has been reported to increase growth, grain yield, resistance to pathogens, and tolerance to abiotic stresses [18,19,20]. For example, Si-treated rice exhibited increased resistance to diseases such as leaf blast, sheath blight, brown spot, and stem rot [21]. Rice is a Si-hyperaccumulating species that absorbs Si in the form of silicic acid (H_4_SiO_4_) from the soil through the roots, and can accumulate Si up to 10% of shoot dry weight [22]. This makes Si an important limiting factor for rice production, and a supply of exogenous Si is necessary for stable rice production systems [23,24].

In previous studies, the contribution of DCY84^T^-treated seeds, Si-coated seeds, and DCY84^T^+Si-treated seeds to rice growth was analyzed by phenotypic observations [25]. At 60 days after sowing (DAS), rice seedlings from seeds treated with DCY84^T^+Si had a higher shoot length and an increased total fresh and dry weight compared to those with DCY84^T^-treated seedlings, Si-coated seedlings, and mock-inoculated seedlings [25]. More interestingly, DCY84^T^-treated seeds or Si-coated seeds did not significantly affect fertility percentage, whereas DCY84^T^+Si had a significantly higher number of spikelets per panicle, and increased grain yield up to 70% than mock-inoculated seedlings [25]. These results support the fact that the increase of plant growth and grain yield was maximized when DCY84^T^ and Si were used together. Although a transcriptome analysis was previously performed by treating Si in rice roots, finding an improvement of suberization and lignification by this treatment [26], further transcriptome analysis on the role of combined DCY84^T^ and Si treatment in early plant growth would be very useful for future applications to enhance crop yield.

In the current study, we performed whole-transcriptome shotgun sequencing (RNA-Seq) analysis on total RNA extracted from roots from control and DCY84^T^+Si-treated seedlings to investigate the effect of the treatment on plant growth. Through this analysis, we identified 576 and 394 genes which were significantly up- and down-regulated, respectively, in response to the combined treatment. Gene ontology (GO) enrichment analysis, MapMan analysis, and analysis of rice genes with known functions were carried out for these candidate genes. Here, we describe and discuss the effect of combined DCY84^T^ and Si treatment which have improved initial growth and increased resistance to environmental stresses.

## 2. Results and Discussion

### 2.1. Combined DCY84^T^ and Si Treatment Stimulates Root Growth of Rice Plants

We compared various traits of rice seedlings grown from untreated seed and from seed subjected to DCY84^T^+Si treatment. Seedlings from treated rice seeds (*Oryza sativa*, *japonica* variety Chilbo), at 3 d after germination, produced plumules and radicles that were significantly longer than those from control seeds (Appendix A). Thereafter, the plants grown from the two types of seeds were monitored for four weeks at intervals of one week (Table 1; Appendix A). A statistical analysis showed that at one week after sowing, plants treated with combined DCY84^T^+Si increased both length and dry weight in root and leaf compared to control plants. However, at two weeks after sowing, all significant differences by the combined treatment disappeared except for root length, which was also lost at three weeks after sowing. On the other hand, the mean root dry weight of the DCY84^T^+Si-treated plants at three weeks after sowing was 20% heavier than that of the control (Table 1). These results are very similar to our previous measurements of root length and dry weight in paddy fields [25]. In addition, rice seedlings at three weeks after sowing consumed most of nutrients in the endosperm of germinating seeds, and more stably adapted to the environment. Therefore, we performed RNA-Seq analysis by sampling the control and DCY84^T^+Si-treated roots from seedlings at 21 DAS.

### 2.2. RNA-Seq Analysis Identified Global Candidate Genes Associated with DCY84^T^+Si Treatment

To identify the genes showing differential expression patterns in response to DCY84^T^+Si treatment, we compared three-week-old roots from treated seedlings with those from untreated control seedlings, with three biological replicates. RNA-Seq revealed that 576 genes were upregulated, and 394 genes were downregulated in the treated plants relative to the corresponding genes in the controls (*p*-values < 0.05 and log_2_ fold-changes > 2; Figure 1A). A heatmap was constructed with data for log_2_ fold-change values for both treated plants and control plants, along with log_2_ intensities in the two sample types of the 970 differentially-expressed genes (DEGs), including both the upregulated and downregulated genes (Figure 1A; Appendix A). In addition, we compared RNA-seq data with the previous data (GSE23723) that performed a microarray with mock and Si-treated roots in rice. We identified 147 genes (log_2_ fold-changes > 2; Appendix A) that were upregulated in Si treatment, and compared them with 576 genes that were upregulated in DCY84^T^+Si treatment. Interestingly, it was found that only three genes overlap between the two groups. These results indicate that the genes, in response to Si treatment and those in response to DCY84^T^+Si treatment, might be quite different.

To verify the DEGs, we selected 6 genes that are upregulated and 6 that are downregulated when treated with DCY84^T^+Si (Appendix A). Then, we checked the expression patterns of 12 genes by real-time quantitative PCR (qPCR). As a result, we confirmed that the RNA-Seq data showed significant positive correlation with qPCR data (Figure 1B). Notably, transcriptional factors or transporter with confirmed expression patterns by qRT-PCR might be primary targets to increase biomass mediated by DCY84^T^+Si treatment.

### 2.3. GO Enrichment Analysis Revealed the Significant Biological Processes Associated with DCY84^T^+Si Treatment

To identify the biological functions of the 576 genes upregulated by DCY84^T^+Si treatment, we performed a GO term enrichment analysis of those genes in the “biological process” category. In all, ten GO terms were highly over-represented in our gene list, with *p*-values < 0.01 and fold-enrichment values of >2-fold (Figure 2; Appendix A), as previously reported [27]. They included biological processes relating to the phenylpropanoid metabolic process (11.76-fold enrichment), L-phenylalanine catabolic process (11.23), L-phenylalanine biosynthetic process (11.23), cellular amino acid and derivative metabolic process (10.29), carboxylic acid metabolic process (8.95), gibberellin metabolic process (8.24), glutamine metabolic process (8.23), protein-chromophore linkage (6.34), oligopeptide transport (2.83), and response to stress (2.42).

Of these, the phenylpropanoid metabolic process was significantly enriched by DCY84^T^+Si treatment (Figure 2). Phenylpropanoids are well known as defense chemicals that protect plants against pathogen attacks through induced systemic resistance (ISR) mediated by PGPB. For example, when *Burkholderia* sp., a member of the PGPB group, was treated with *Vitis vinifera*, phenolic compounds accumulated and the cell walls of the exodermis and cortical cell layers were strengthened, increasing disease resistance [28]. In addition, when root-inducing T-DNA peas are treated with the endophytic bacterium *B. pumilus*, the pea root-rotting fungus *Fusarium oxysporum* forms wall appositions containing phenolic compounds to limit pathogen growth [29]. Finally, treatment of *B. pumilus* with tomato plants inoculated with the vascular fungus *F. oxysporum* reduces the growth of pathogens through the formation of wall appositions composed of phenolic compounds [30]. Interestingly, rice treated with DCY84^T^ is more resistant to *Xanthomonas oryzae pv. oryzae* (*Xoo*) compared with controls [16]. These results suggest that the synthesis of phenylpropanoid may also occur in rice roots treated with DCY84^T^+Si.

Then, the L-phenylalanine catabolic and L-phenylalanine biosynthetic processes were significantly enriched. As an aromatic amino acid, phenylalanine is a central molecule in plant metabolism, and its biosynthetic pathways and regulation have been extensively studied [31]. Phenylalanine acts as a precursor to a range of phenolic secondary metabolites, including phenylpropanoids, flavonoids, and lignin [32]. There have been several reports that salt stress stimulates the phenylpropanoid biosynthetic pathway to produce various phenolic compounds with antioxidant ability. For example, plants of *Thymus vulgaris* and *Thymus daenensis* grown under salt stress exhibited increased antioxidant activity as a result of the increased production of phenolic compounds [33]. In addition, the transformation of *Arabidopsis thaliana* with the grape bHLH transcription factor gene, *VvbHLH1*, which regulates flavonoid biosynthesis, increased the accumulation of flavonoids and enhanced salt tolerance in the transgenic plant [34]. In a previous study, DCY84^T^-treated rice improved plant growth in saline coastal soils [16]. These results suggest that treatment with DCY84^T^+Si may increase the production of phenylpropanoids or other phenolic compounds through stimulation of the phenylpropanoid biosynthetic pathway.

### 2.4. MapMan Analysis Revealed Involvement of DCY84T+Si Treatment in Phenylpropanoid Metabolism and Ethylene Regulation

The MapMan program is an effective tool for visualizing diverse overviews associated with high-throughput transcriptomics data [35]. We uploaded the fold-change data and locus IDs of the 576 upregulated genes to various overviews installed in the MapMan program (Figure 3; Appendix A). In the metabolism overview, we identified six genes upregulated in “Phenylpropanoids & Phenolics” metabolism belonging to secondary metabolism (blue box in Figure 3A). As we found that the phenylpropanoid metabolic process was the biological process which is most enriched by GO enrichment analysis in Section 2.3, it was predictable that many phenylpropanoid-associated genes would be identified in the metabolism overview.

We also identified 13 upregulated genes in the ethylene regulation pathway using a regulation overview (blue box in Figure 3B). Rhizobacteria, which include most PGPBs, colonize the roots of growing plants and react, using host-derived chemicals in root exudates, and using tryptophan to synthesize the natural auxin indole-3-acetic acid (IAA) [36]. Both bacterial IAA and endogenous plant IAA can stimulate plant growth and induce the synthesis of aminocyclopropane-1-carboxylate (ACC) synthase, one of the ethylene biosynthesis pathway enzymes in plants, where it plays a role in converting S-adenosyl methionine to ACC, the immediate precursor of ethylene. Some of the ACC molecules synthesized are converted to ammonia and α-ketobutyrate by ACC deaminase in bacteria [9,37,38,39,40]. Thus, the activity of bacterial ACC deaminase decreases the concentration of ethylene produced in plants, which results in increased plant growth [38]. We observed that the root length and dry weight of plants treated with DCY84^T^+Si increased significantly compared to the control (Table 1; Appendix A). These results suggest that DCY84^T^ with Si treatment might be a PGPB capable of synthesizing ACC deaminase.

### 2.5. Functions of Candidate Genes Associated with DCY84^T^+Si Were Evaluated through Literature Searches

To further evaluate the functional significance of our candidate DEGs, we identified the known functions for these DEGs from previous studies. Of the DEGs, we found 30 genes upregulated by DCY84^T^+Si treatment with known functions (Table 2). Twelve of the genes were related to responses to various abiotic stresses, namely, *drought-responsive ERF 1* (*OsDERF1*; [39]) and *basic helix–loop–helix domain148* (*OsbHLH148*; [40]) for drought; *ERF protein associated with tillering and panicle branching* (*OsEATB*; [41]) and *SALT-RESPONSIVE ERF1* (*SERF1*; [42]) for salinity; *OsWRKY76* [43] for cold; *zinc finger protein252* (*ZFP252*; [44]) and *Ca^2+^-dependent protein kinase 4* (*OsCPK4*; [45]) for drought and salinity; *trehalose-6-phosphate phosphatase1* (*OsTPP1*; [46]) for salinity and cold; *mitogen-activated protein kinase5* (*OsMAPK5*; [47]), *dehydration-responsive element-binding transcription factor 1F* (*OsDREB1F*; [48]), and *OsDREB1C* [49] for drought, salinity, and cold; and *basic helix*–*loop*–*helix 133* (*OsbHLH133*; [50]) for tolerance to other soil stresses. With respect to biotic stresses, the functions of seven genes were identified: *fatty acid desaturase7* (*OsFAD7*; [51]), *1-aminocyclopropane-1-carboxylic acid synthase 2* (*OsACS2*; [52,53]), *OsMAPK5* [47], and *BROAD-SPECTRUM RESISTANCE 1* (*BSR1*; [54]) for rice blast resistance; *OsWRKY71* [55] for bacterial blight resistance; and *OsWRKY28* [56,57] and *OsWRKY76* [43,56] for both blast and bacterial blight resistance. Nine genes were related to morphological traits, namely, *Elicitor 5* (*EL5*; [58]), *tryptophan deficient dwarf 1* (*tdd1*; [59]), and *cZ-O-glucosyltransferase 2* (*cZOGT2*; [60]) for root traits; *response to exogenous JA 1* (*RERJ1*; [61]), *cytochrome P450 monooxygenase 734A4* (*CYP734A4*; [62]), and *OsCPK4* [45] for dwarf habit; *BRASSINOSTEROID UPREGULATED1* (*bu1*; [63]) for leaf and seed change (grain size); and *dense and erect panicle 3* (*dep3*; [64]) and *ERF protein associated with tillering and panicle branching* (*OsEATB*; [41]) for panicle development. Two of the genes were related to physiological traits, including *DEFECT IN EARLY EMBRYO SAC1* (*OsDEES1*; [65]) for sterility and *OsACS2* [66] for spikelet fertility.

As expected from earlier results from the current study (Table 1; Appendix A), several DEGs were found among the known genes associated with root biomass traits (Table 2). In addition, most of the known DEG genes are more likely associated with resistance- or tolerance-related traits (Table 2). These findings indicate that our candidate genes are potentially involved in plant responses to abiotic stresses, including drought, salinity, and cold, as well as growth promotion. Previous literature had reported that *A. thaliana* treated with DCY84^T^ was more tolerant of drought, salinity, and aluminum treatments [16]. Our findings and the previous study support the hypothesis that rice plants treated with DCY84^T^+Si can better withstand various abiotic stresses such as drought or cold. Furthermore, uncharacterized DEGs might be useful targets for further study to enhance abiotic stress tolerance or growth promotion.

### 2.6. Treatment with DCY84^T^+Si Resulted in Increased Rice Drought Tolerance

We had predicted the positive effects of DCY84^T^+Si treatment on root biomass, biotic stress resistance, and abiotic stress tolerance through GO enrichment analysis, MapMan analysis, and analysis of rice DEGs with known functions. Of these, tolerance of abiotic stresses, including drought stress, was hypothesized to be significantly affected by DCY84^T^+Si. To test this, we conducted the following experiment to determine whether rice plants treated with DCY84^T^+Si showed greater tolerance to drought stress. Control and DCY84^T^+Si-treated plants were grown for four weeks and then subjected to drought stress for five days, after which the plants were allowed to recover for 10 days (Figure 4A). Plants treated with DCY84^T^+Si showed significantly greater tolerance to drought stress than the control plants. The expression patterns were then examined for two genes that had been identified as molecular markers of the drought-stress response, i.e., *OsDREB2b* (*LOC_Os05g27930*) and *OsbZIP23* (*LOC_Os02g52780*) [67]. As expected, stressed roots (from Day 5) showed increased expression of those genes, supporting the hypothesis that samples collected under drought stress were well qualified for further analysis (Figure 4B). Of the known genes, seven upregulated genes related to drought stress were selected (Table 2), and their expression patterns in treated and control plants were compared. All seven genes were found to be expressed at a significantly higher level in plants treated with DCY84^T^+Si than in control plants (Figure 4C). These results suggest that treatment with DCY84^T^+Si increased the expression of several drought-related genes, resulting in increased drought tolerance being conferred upon the treated rice plants.

### 2.7. Analyses of Predicted Protein–Protein Interactions Associated with DCY84^T^+Si Treatment Suggest a Regulatory Model

Regulatory genes, such as those encoding transcription factors, are primary targets when investigating diverse abiotic stress responses and developmental processes. Understanding the regulatory relationship among upregulated genes can lead to a new strategy for increasing tolerance to environmental stress as a result of DCY84^T^+Si treatment. We utilized the Rice Interactions Viewer to generate a hypothetical protein–protein interaction network associated with the 576 genes upregulated in response to DCY84^T^+Si treatment [68]. We then refined the network by using genes in the following four categories as the query: 13 transcription factors (pink circles, Figure 5), one transporter (yellow circle), four kinases (green circles), and two genes functionally characterized to be associated with abiotic stress tolerance (blue circles, Figure 5). 

We predicted that OsMAPK5 interacts with six TFs and one transporter in the network (Figure 5). According to previous studies, the *OsMAPK5* gene, its protein, and its kinase activity were all reported to be induced by abscisic acid and various biotic and abiotic stresses [47]. Interestingly, the expression of *OsMAPK5* was also shown to be altered by some PGPBs. For example, greenhouse-grown rice treated with *Bacillus amyloliquefaciens* NBRISN13 showed decreased expression of *OsMAPK5* in the leaves [69]. In contrast to this result, we found that the expression of *OsMAPK5* was increased in the rice root when treated with DCY84^T^+Si (Figure 4B). Xiong et al. [47] reported that *OsMAPK5*-overexpressed lines not only increased kinase activity, but also increased tolerance to drought, salt, and cold stresses. These results do not provide information on the pathway by which DCY84^T^+Si induces *OsMAPK5* expression, but our network model suggests the possible regulatory pathways associated with OsMAPK5 which was stimulated in response to DCY84^T^+Si treatment. Further functional analysis using our transcriptional regulation model might shed light on the OsMAPK5-mediated signaling and transcriptional regulation pathway.

## 3. Materials and Methods

### 3.1. Plant Materials and Phenotypic Observation

The *Paenibacillus yonginensis* strain DCY84^T^ was grown at 30 °C on trypticase soy broth for 16 h. The culture broth was centrifuged, and the pelleted cells were resuspended in a dilute saline solution (0.85% NaCl). Surface-sterilized seeds of *O. sativa, japonica* cv. Chilbo (Rural Development Administration, Jeonju, Korea) were treated by soaking them in the bacterial suspension (10^8^ CFU ml^−1^) or saline solution (control) for 30 min. The binder solution was composed of 10% Na_2_SiO_3_, 5% humic acid, 3% sodium alginate, 0.05% molybdenum, and 0.01% carboxymethyl chitosan. 500 g of Zeolite was mixed with 2 kg of inoculated rice seeds, and 300 mL of binder solution was sprayed [25,70]. For the control seeds, only a binder was added. Treated and control seeds were allowed to germinate by imbibing them in water for three days and then placing the seedlings in a growth incubator (Younghwa Science, Daegu, Korea) for four weeks (14-h light/10-h dark, 28 °C/22 °C). To observe the effects of DCY84^T^+Si on the growth of rice, the length and dry weight of the roots of control and treated plants were measured at seven-day intervals for four weeks (Table 1; Appendix A).

### 3.2. Statistical Analyses

Thirty plants were used at each stage to determine the lengths and dry weight of roots and leaves. All data were presented as mean ± standard deviation. Statistical analyses were performed using the Student’s *t* test (* *p <* 0.05; ** *p* < 0.01; and *** *p* < 0.001). Then, we sampled the roots of the control and DCY84^T^+Si-treated plants grown for three weeks for RNA-Seq analysis.

### 3.3. RNA-Seq Analysis

We used the Illumina platform to generate sequences (approximately 26 GB) of three independent total RNA samples from the roots of each of the DCY84^T^+Si-treated and control seedlings. In each transcriptome sample, 100-bp paired-end sequences were assessed with a FastQC toolkit [71]. Any adapter contaminations or low-quality sequences (pPhred +33 and -q 20) were removed using both Cutadapt [72] and its wrapper tool, Trimgalore [73]. The resultant high-quality sequences were used for our TopHat2 pipeline [74]. On average, 94% of the filtered sequences were mapped to the International Rice Genome Sequencing Project (IRGSP) 1.0 reference genome [75], and the gene features were estimated based on the gff3 annotation file from the Rice Genome Annotation Project (RGAP) database (http://rice.plantbiology.msu.edu) [76]. DEGs were evaluated using Cuffdiff to compare treatment conditions. Genes with *p*-values < 0.05 and log_2_ fold-changes > 2 (i.e., fold-change > 4) were considered to be differentially expressed. Further screening among the initial DEGs was based on fragments per kilobase per million fragments mapped (FPKM) values [77]. The selected DEGs created a heatmap using the Multi Experiment Viewer (MeV_4-9-0) software tool [78,79].

### 3.4. GO Enrichment Analysis

We employed the GO enrichment tool [80] to determine the biological roles of selected genes listed in the Rice Oligonucleotide Array Database (http://ricephylogenomics-khu.org/road/go_analysis.php). This included any genes that were upregulated during DCY84^T^+Si treatment. An enrichment value higher than standard (1) meant that the selected GO term was over-represented. Terms with >3-fold enrichment values were also considered.

### 3.5. MapMan Analysis

The rice MapMan classification system covers 36 BINs, each of which could be extended in a hierarchical manner into subBINs [67,81,82]. Using diverse MapMan tools, a significant gene list selected from high-throughput data analysis can be integrated to produce diverse overviews. Here, we generated a dataset carrying locus IDs from the RGAP annotation version 7.0 in addition to average log_2_ fold-change data for controls versus DCY84^T^+Si conditions. To describe the functional classification of genes upregulated in response to DCY84^T^+Si, we used two overviews: metabolism and regulation.

### 3.6. Analysis of Rice Genes with Known Functions

To evaluate the functional significance of our candidate genes, we compared our gene list with the Overview of Functionally-Characterized Genes in Rice Online database (http://qtaro.abr.affrc.go.jp/ogro), which summarizes rice genes with known functions [83], and then grouped them into three major categories: resistance- or tolerance-related trait, morphological trait, and physiological trait.

### 3.7. Drought Stress Treatment

Plants of *japonica* rice (*Oryza sativa*) cv. Chilbo from either control seeds or seeds dressed with DCY84T+Si were allowed to germinate by imbibing them in water for three days and then placing the seedlings in a growth incubator (Younghwa Science, Daegu, Korea) for four weeks (14-h light/10-h dark, 28 °C (day)/22 °C (night), humidity 80%). On average, 800 g of dried soil was used to grow ten plants (five control plants vs. five DCY84^T^+Si plants) in each pot. Ten replicate pots were used for phenotypic observation. Drought stress was applied to four-week-old seedlings by withholding irrigation for 5 d (14-h light/10-h dark, 28 °C (day)/22 °C (night), humidity 40%), after which the seedlings were rewatered for 10 d (14-h light/10-h dark, 28 °C (day)/22 °C (night), humidity 80%). Plant phenotypes were photographed with a camera (Canon EOS 550D; Canon, Tokyo, Japan) before and after drought stress treatment and during the recovery phase.

### 3.8. RNA Extraction and Quantitative RT-PCR (qRT-PCR) Anayslis

Three independent replicates (three plants per replicate) of each of the three-week-old control roots and DCY84^T^+Si-treated roots were sampled and immediately frozen in liquid nitrogen. Total RNA was extracted using RNAiso Plus according to the manufacturer’s protocol (Takara Bio, Kyoto, Japan). The first-strand cDNA was synthesized with MMLV Reverse Transcriptase (Promega, Fitchburg, WI, USA) and the oligo(dT)-15 primer. The qPCR was performed by Qiagen Rotor-Gene Q real-time PCR cycler using the following thermal cycling procedure: 95 °C for 10 s, 60 °C for 30 s, and 72 °C for 1 min. To normalize the amplified transcripts, we used a primer pair for rice ubiquitin 5 (*OsUbi5/Os01g22490*) [84]. Finally, a Student’s t-test was used for statistical analysis. All primers for the genes used in these analyses are presented in Appendix A.

## 4. Conclusions

We demonstrated that a combination DCY84^T^+Si treatment increases root growth in the seedling stage, and increases tolerance of drought stress. GO analysis, MapMan, and literature searches indicated that the tolerance response might be due to the upregulation of genes involved in the phenylpropanoid and ethylene metabolic pathways. Gene indexed mutants are available for more than half of the rice genome with a well-established gene editing system [84]. With the benefit for further research as a model crop plant, understanding and exploiting the interactions between bacteria and plants can be a powerful tool by which to enhance agronomic traits such as nutrient use efficiency, crop yield, and stress tolerance.

## Figures and Tables

**Figure 1 ijms-20-05883-f001:**
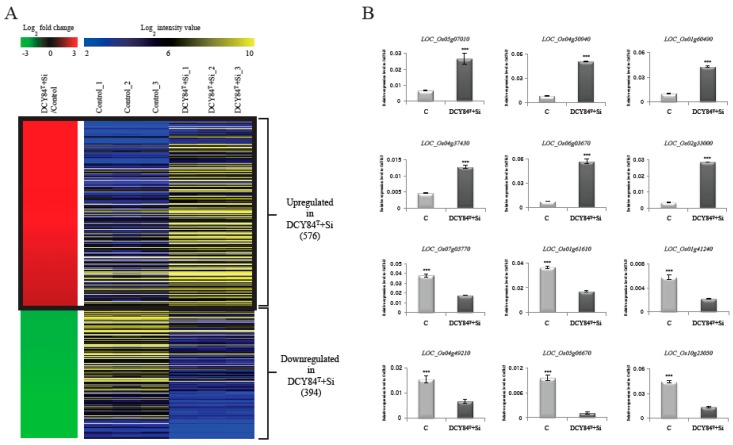
Heat map of differentially-expressed genes during DCY84^T^+Si treatment. Using RNA-seq data analysis under criteria of FPKM > 4, *p*-values < 0.05, and log_2_ ratio of < −2 for > 2 of DCY84^T^+Si-treated plant roots versus normal plant roots (control), we identified 1455 differentially-expressed genes (**A**). In the left panel, red indicates upregulation in DCY84^T^+Si/control comparisons; green indicates downregulation in DCY84^T^+Si/control comparisons. The right panel shows average normalized log_2_ FPKM values from RNA-seq experiments; blue indicates the lowest expression level, and yellow the highest. The effects of DCY84^T^+Si were checked by monitoring expression patterns of 12 genes (**B**). The y axis indicates expression level relative to *OsUBI5/Os01g22490* (internal control); the x axis indicates samples used for qRT-PCR. *** *p* < 0.001. Detailed data about RNA-seq analysis are presented in Appendix A.

**Figure 2 ijms-20-05883-f002:**
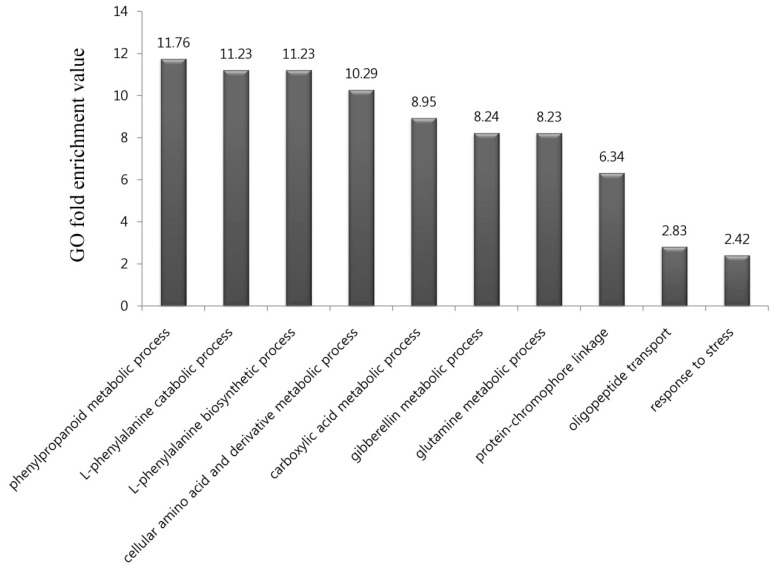
Gene ontology (GO) enrichment analysis in the “biological process” category for genes upregulated and downregulated in response to DCY84^T^+Si treatment. In all, 10 GO terms were over-represented under > two-fold enrichment value, with *p*-values < 0.01. Details of GO assignments are presented in Appendix A.

**Figure 3 ijms-20-05883-f003:**
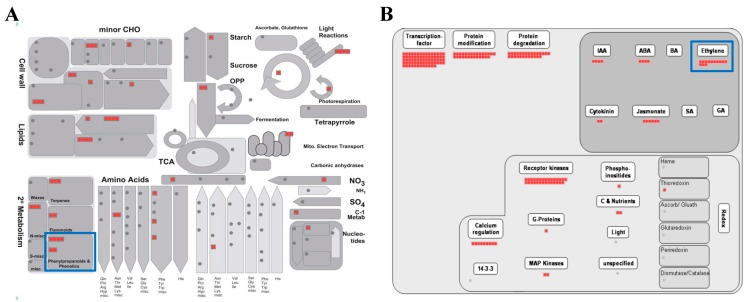
MapMan analysis of genes associated with a response to DCY84^T^+Si treatment. Overviews: (**A**) metabolism-response overview, and (**B**) regulation overview. Red boxes indicate genes upregulated by DCY84^T^+Si. Detailed information is presented in Appendix A.

**Figure 4 ijms-20-05883-f004:**
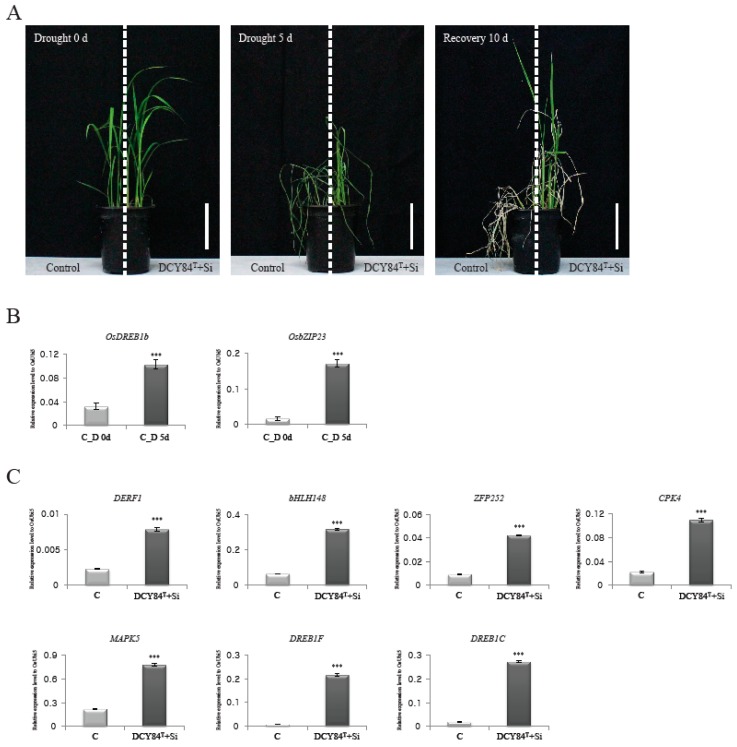
Drought-stress response mediated by DCY84^T^+Si. Control and DCY84^T^+Si treated plants grown in plastic pots for 4 weeks was exposed to drought stress for 5 d. Photo were taken 10 d after re-watering (**A**). Effects of water deficiency were checked by monitoring expression patterns of drought-stress marker genes, *OsDREB1b* and *OsZIP23* (**B**). Analyses of transcripts of *DERF1*, *bHLH148*, *ZFP252*, *CPK4*, *MAPK5*, *DREB1F*, and *DREB1C* for control and DCY84T+Si (**C**). The expression levels were normalized to that of *OsUBI5/Os01g22490* (internal control) using real-time PCR analysis. C, control without drought treatment; D 0d, Drought stress treatment for 0 day; D 5d, Drought stress treatment for 5 days. Scale bar = 10 cm. N = 3 (**A**). *** *p* < 0.001 (**B**,**C**).

**Figure 5 ijms-20-05883-f005:**
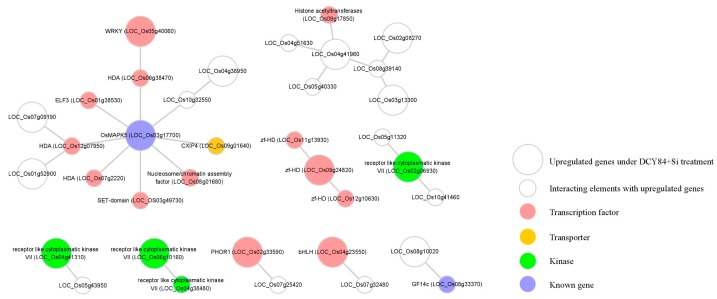
Construction of regulatory network associated with genes upregulated under DCY84^T^+Si treatment. Using Rice Interaction Viewer and Cytoscape tools, we queried the predicted protein–protein interaction network associated with 13 transcription factors (pink circles), one transporter (yellow circle), four kinase (green circles), and two functionally-characterized genes in environmental stress (blue circles). The large circle represents the upregulated gene under DCY84^T^+Si treatment. The small circle represents an interacting element with upregulated genes.

**Table 1 ijms-20-05883-t001:** Comparison of the length and dry weight of the leaves and roots between normal conditions and both DCY84^T^ and Si treatments.

Length (cm)	Root		Leaf		Dry Weight (mg)	Root		Leaf	
	Chilbo	Chilbo _*Py* + Si	Chilbo	Chilbo _*Py* + Si		Chilbo	Chilbo _*Py* + Si	Chilbo	Chilbo _*Py* + Si
One week	7.16 ± 1.02	9.07 ± 1.34 **	11.99 ± 0.97	13.78 ± 1.31 ***	One week	2.08 ± 0.38	2.56 ± 0.31 **	7.82 ± 0.59	9.5 ± 0.78 ***
Two weeks	9.39 ± 1.71	13.04 ± 2.05 *	26.48 ± 1.91	25.52 ± 1.83	Two weeks	7.04 ± 0.95	7.89 ± 2.01	38.34 ± 3.44	37.73 ± 4.78
Three weeks	15.12 ± 1.24	15.05 ± 1.59	36.52 ± 1.77	36.21 ± 2.84	Three weeks	35.50 ± 2.15	42.72 ± 5.12 **	179.23 ± 15.52	176.98 ± 18.67
Four weeks	16.11 ± 1.03	16.32 ± 1.14	42.39 ± 2.55	43.12 ± 1.93	Four weeks	60.56 ± 14.43	62.41 ± 15.87	255.62 ± 33.85	260.23 ± 37.79

* Statistical significance was assigned at *p* < 0.05; ** Statistical significance was assigned at *p* < 0.01; and *** Statistical significance was assigned at *p* < 0.001.

**Table 2 ijms-20-05883-t002:** Summary of functionally-characterized genes through literature searches associated with DCY84^T^ and Si.

Major Category ^a^	Minor Category ^b^	Locus_Id ^c^	Gene Name	Gene Symbol	Method ^d^	DOI References ^e^
RT ^f^	Drought tolerance	LOC_Os08g35240	drought-responsive ERF 1	OsDERF1	KD, OX	10.1371/journal.pone.0025216
RT	Drought tolerance	LOC_Os03g53020	basic helix-loop-helix domain148	OsbHLH148	OX	10.1111/j.1365-313X.2010.04477.x
RT	Salinity tolerance	LOC_Os09g28440	ERF protein associated with tillering and panicle branching	OsEATB	OX	10.1104/pp.111.179945
RT	Salinity tolerance	LOC_Os05g34730	SALT-RESPONSIVE ERF1	SERF1	M	10.1105/tpc.113.113068
RT	Cold tolerance	LOC_Os09g25060	OsWRKY76	OsWRKY76	OX	10.1093/jxb/ert298
RT	Drought, salinity tolerance	LOC_Os12g39400	zinc finger protein252	ZFP252	OX	10.1016/j.febslet.2008.02.052
RT	Drought, salinity tolerance	LOC_Os02g03410	Ca2+-dependent protein kinase 4	OsCPK4	OX	10.1104/pp.113.230268
RT	Salinity, cold tolerance	LOC_Os02g44230	trehalose-6-phosphate phosphatase1	OsTPP1	OX	10.1007/s00425-008-0729-x
RT	Drought, salinity, cold tolerance	LOC_Os03g17700	Mitogen-activated protein kinase5	OsMAPK5	KD, OX	10.1105/tpc.008714
RT	Drought, salinity, cold tolerance	LOC_Os01g73770	dehydration-responsive element-binding transcription factor 1F	OsDREB1F	OX	10.1007/s11103-008-9340-6
RT	Drought, salinity, cold tolerance	LOC_Os06g03670	dehydration-responsive element-binding transcription factor 1C	OsDREB1C	OX	10.1093/pcp/pci230
RT	Other soil stress tolerance	LOC_Os12g32400	basic helix loop helix 133	OsbHLH133	M	10.1111/j.1365-3040.2012.02569.x
RT	Blast resistance	LOC_Os03g18070	fatty acid desaturase7	OsFAD7	KD	10.1093/pcp/pcm107
RT	Blast resistance	LOC_Os04g48850	1-aminocyclopropane-1-carboxylic acid synthase 2	OsACS2	KD, OX	10.1104/pp.110.16241210.1111/pbi.12004
RT	Blast resistance	LOC_Os03g17700	Mitogen-activated protein kinase5	OsMAPK5	KD, OX	10.1105/tpc.008714
RT	Blast resistance	LOC_Os09g36320	BROAD-SPECTRUM RESISTANCE 1	BSR1	OX	10.1111/j.1467-7652.2010.00568.x
RT	Bacterial blight resistance	LOC_Os02g08440	OsWRKY71	OsWRKY71	OX	10.1016/j.jplph.2006.07.006
RT	Blast, bacterial blight resistance	LOC_Os06g44010	OsWRKY28	OsWRKY28	OX	10.1007-s11103-013-0032-5 10.1007/s12284-010-9039-6
RT	Blast, bacterial blight resistance	LOC_Os09g25060	OsWRKY76	OsWRKY76	OX	10.1093/jxb/ert29810.1007/s12284-010-9039-6
MT ^g^	Root	LOC_Os02g35347	Elicitor 5	EL5	Others	10.1111/j.1365-313X.2007.03120.x
MT	Root	LOC_Os04g38950	tryptophan deficient dwarf 1	tdd1	M	10.1111/j.1365-313X.2009.03952.x
MT	Root	LOC_Os04g46990	cZ-O-glucosyltransferase 2	cZOGT2	OX	10.1104/pp.112.196733
MT	Dwarf	LOC_Os04g23550	response to exogenous JA 1	RERJ1	KD, OX	10.1016/j.bbrc.2004.10.126
MT	Dwarf	LOC_Os06g39880	cytochrome P450 monooxygenase 734A4	CYP734A4	OX	10.1111/j.1365-313X.2011.04567.x
MT	Dwarf	LOC_Os02g03410	Ca2+-dependent protein kinase 4	OsCPK4	KD	10.1104/pp.113.230268
MT	Leaf, Seed	LOC_Os06g12210	BRASSINOSTEROID UPREGULATED1	bu1	OX	10.1104/pp.109.140806
MT	Panicle flower	LOC_Os06g46350	dense and erect panicle 3	dep3	M	10.1007/s00122-011-1543-6
MT	Panicle flower	LOC_Os09g28440	ERF protein associated with tillering and panicle branching	OsEATB	OX	10.1104/pp.111.179945
PT ^h^	Sterility	LOC_Os09g38850	DEFECT IN EARLY EMBRYO SAC1	OsDEES1	KD	10.1104/pp.112.203943
PT	Spikelet fertility	LOC_Os04g48850	1-amino-cyclopropane-1-carboxylate (ACC) synthase 2	OsACS2	KD	10.1111/tpj.12508

^a^ Of agronomic traits associated with functionally-characterized genes out of candidate genes in this study. ^b^ indicates sub-agronomic trait categories in each of major categories. ^c^ indicates the systematic locus identifiers used in the MSU rice database. ^d^ indicates the methods used for the functional characterization: M indicates mutants by T-DNA/Tos17/Ds insertion; KD, knockdown mutants by RNAi or anti-sense approaches; OX, overexpressed mutants by transgenic approaches; and others, those by other methods besides three major methods. ^e^ indicates Digital Object Identifier (DOI). ^f^ indicates Resistance or Tolerance relating trait. ^g^ indicates Morphological trait. ^h^ indicates Physiological trait.

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
