# Peer review of "Genome-Wide Transcriptome Analysis of Rice Seedlings after Seed Dressing with Paenibacillus yonginensis DCY84T and Silicon"

_ijms, 2019, doi:10.3390/ijms20235883_

Round 1
Reviewer 1 Report
Yoo et al. have improved the manuscript accordingly. But I still have some concern and comments on it.
Due to the limited budget, the authors did not examine the transcriptome of seedlings treated with silicon or DCY84 alone. But I think the authors could select a few candidate genes which might be involved in the promotion of plant growth by the treatments and examined their expression under 4 different treatment conditions. This might help to partially elucidate the effects of different treatments on candidate gene expression. On the other hand, the authors mentioned that the transcriptome analysis of Si-treated rice was done before. The authors may find out the effect of Si and the interaction of Si and DCY84 through comparing their data with the previous one. In section 2.2, I think it is not necessary to name out all the genes they tested. The authors might list them in the supplementary table and briefly mention the category of gene function. Based on the root phenotype shown in supplementary figure 1B and the statement in conclusion “combination DCY84T+Si treatment promotes seed germination”, did seeds treated with Si+DCY84 germinate earlier? If they did, it is reasonable to see the increase of shoot and root growth under treatment conditions only at the beginning.
Author Response
Responses to reviewer’s comments
Comment 1: Due to the limited budget, the authors did not examine the transcriptome of seedlings treated with silicon or DCY84 alone. But I think the authors could select a few candidate genes which might be involved in the promotion of plant growth by the treatments and examined their expression under 4 different treatment conditions. This might help to partially elucidate the effects of different treatments on candidate gene expression.
Response 1: We think it is a good idea to investigate the expression of several candidate genes under 4 different treatment conditions (control, DCY84, silicon, DCY84 + silicon). However, 10 days is not enough to obtain root samples for 4 different treatment conditions and investigate gene expression because we have limited time (10 days) for the revision. If journal allows more time for the revision, we will try and otherwise, it will be difficult to follow this comment. As we found the significant difference between DCY84+Si treatment and Si treatment from the comparative transcriptome analysis, the novelty of our research can be argued from this point of view with enhanced growth promotion of the combined treatment.
Comment 2: On the other hand, the authors mentioned that the transcriptome analysis of Si-treated rice was done before. The authors may find out the effect of Si and the interaction of Si and DCY84 through comparing their data with the previous one.
Response 2: We compared Si treated transcriptome data with DCY84+Si treated RNA-seq data. As a result, we found a few overlap (three genes) between two experiments which indicate that the effect of DCY84+Si treatment might be quite different with that of Si treatment. Thus, enhanced growth promotion can be more effectively explained with combined treatment. Regarding this, we have written the contents in manuscript (lines 111~117) and added the Table S2 which lists the differentially expressed genes between Si treatment and control.
Comment 3: In section 2.2, I think it is not necessary to name out all the genes they tested. The authors might list them in the supplementary table and briefly mention the category of gene function.
Response 3: All gene information in section 2.2 is listed in Table S3. Explanation for each gene has been removed from main manuscript (line 118~121).
Comment 4: Based on the root phenotype shown in supplementary figure 1B and the statement in conclusion “combination DCY84T+Si treatment promotes seed germination”, did seeds treated with Si+DCY84 germinate earlier? If they did, it is reasonable to see the increase of shoot and root growth under treatment conditions only at the beginning.
Response 4: We did not identify the earlier germination in the DCY84T+Si treatment. However, obviously, at 3 days after germination, the root growth of the plants increased more than the control. Thus, we modified the “combination DCY84T+Si treatment increases root growth in the seedling stage” instead of the “combination DCY84T+Si treatment promotes seed germination” (line 381~382).
This manuscript has not been published or presented elsewhere in part or in its entirety and is not under consideration by another journal. All the authors have approved the manuscript and agree with submission to your journal. We have no conflicts of interest to declare.
Thank you for your consideration. I look forward to hearing from you.
Sincerely,
Ki-Hong Jung
Professor
Graduate School of Biotechnology
Kyung Hee University, Yongin 17104, Korea
Reviewer 2 Report
The authors have addressed my concerns about the previous version. I am fine with it being published now.
Author Response
Thank you for your consideration.
Sincerely,
Round 2
Reviewer 1 Report
The revised version has improved. I only have one more suggestion. Since the authors did statistic analysis for analyzing growth promoting effects, the methods they used need to be described in the material and methods.
Author Response
Responses to reviewer’s comments
Comment: The revised version has improved. I only have one more suggestion. Since the authors did statistical analysis for analyzing growth promoting effects, the methods they used need to be described in the material and methods.
Response: We have written the statistical analysis contents in Materials and Methods (line 323~327).
This manuscript is a resubmission of an earlier submission. The following is a list of the peer review reports and author responses from that submission.
Round 1
Reviewer 1 Report
Review
The article “Genome-wide Transcriptome Analysis of Rice Seedlings after Seed
Dressing with Paenibacillus yonginensis DCY84T and Silicon” by Yo-Han Yoo with co-authors describe the effect of DCY84T and zeolite on rice under normal condition and under drought stress.
The general idea is very interesting. Authors used very modern methods for investigation. But the current manuscript cannot be suggested for publication, because:
There is no new information about mechanisms and effect. These types of microorganisms and zeolites are tested many times and demonstrated the effect on drought resistance. Zeolite is not only Si. There are numerous elements is present in zeolites. Therefore we cant’ say that this is Si effect without any additional chemical tests (for example total Si in plant tissue) ; Considering that microorganisms and Si (zeolites) has different mechanisms of effect on plant, generally the following treatments must be realized: control, Microorganisms, Si(zeolite), Microorganisms+ Si (zeolites). But not only control and Microorganisms+ Si (zeolites). The discussion contain only general phrases and literature review (which must be in Introduction).
There is some small remarks for authors.
Introduction
Page 1 line 39
The citation #5 is declared about phytoremediation, but not alternative for chemical fertilizer. In basic, for plant grow needs nitrogen, potassium, phosphorus and microelements. I am not shore that ONLY one type of bacteria can to be complete alternative for plant nutrition….Please or remove such strong suggestion or give data how PGPB provide phosphorus, potassium…especially that next sentence e completely refute previous sentence.
Materials and Methods
Materials and methods is important chapter of any article, because any person, which read article must have ability to repeat the experiment. Therefore all details must be present in manuscript.
What is mean “saline solution”? What is salt? Concentration?
Zeolite. There is many types of zeolites. Please give more information – chemical composition, source. Size of zeolite particles?
Drought stress
How you verify that plant had drought stress? It must be special test, which show it. Without such test is not possible to verify it.
Result
The figure 1 and table 1 show same effect. The figure 1 is not important in this manuscript.
Reviewer 2 Report
In this manuscrip, Yoo et al. compared the gene expression profile of rice seedlings treated with Paenibacillus yonginensis DCY84T and silicon and untreated seedlings. Hundreds of genes were up- and down-regulated in the treated seedlings. GO and metabolic analyses suggested the alteration of phenylalanine metabolic process and phenylpropanoid-related pathway in the treated seedlings might be one of the benefits resulted from the treatment of Paenibacillus yonginensis DCY84T and silicon leading to the enhancement of drought tolerance of rice seedlings. It is important to elucidate the benefits of PGPM or other treatment through genome-wide transcriptomic or proteomic analysis, but I think there are still some points need to be clarified to make this story complete.
The aim of this study is not clear. It seems that the authors tried to find out the key factors which promote the growth and stress resistance of rice seedlings treated with DCY84T and silicon by transcriptomic analysis. But the authors chose treated and untreated plants as the study materials. Any alternation observed might be caused either by PGPB or silicon treatment or the interaction between two treatments. This analysis cannot illustrate the impacts of treatment individually which might limit the application in the future. The difference of root length between treated and untreated plants was only shown 2 weeks after sowing but the difference of root dry weight was shown 3 weeks after sowing. Why did the authors used 21-DAS seedlings for transcriptomic analysis? The shoot/root length and dry weight were shown in table 1. The authors did not do any statistic analysis but claimed the difference observed between treatment and control. I strongly suggest to do the statistic analysis to convince the readers that the difference observed is real. Although the manuscript is not too long, it is not concise. Especially in page 4 and 8, the authors described the potential function of genes they picked for qRT-PCR or related to stress resistance individually which were already shown in the tables. It is hard to tell the points that the authors tried to emphasized in these paragraphs. In the subsection 2.5, the authors tried to link the potential function of these DEGs with the phenotype they observed by literature study. I think this section should be in the discussion instead of results. In the introduction, the authors said “Here, we describe and discuss the potential of rice seeds treated with both DCY84T and Si to reduce the use of chemical fertilizers and to enhance the sustainability and cost–benefit ratio of rice production,” but they did not carry out any experiments to show the benefits of treatments supporting the growth and crop production with low nutrient supply. In the discussion, the authors discussed the IAA production and the phosphate solubilizing activity in PGPBs which did not directly link to this study. On the other hand, the results of this study was not discussed further in this section. It is a bit weird. In the material and methods, the authors did not describe the methods used for RNA isolation and statistic analysis they used for qRT-PCR data analysis.
Reviewer 3 Report
This manuscript investigates changes in rice phenotype and gene expression after joint treatment of seeds with Si and a strain of Paenibacillus yonginensis. Replication was limited, only a single variety was used, and individual effects of treatments were not investigated (see below), all of which limits the inferences that can be made. On the other hand, it is a thorough study that makes its main points convincingly, and the English is adequate. The prominence of changes in expression of genes associated with phenylalanine and ethylene metabolism was an interesting result, and the confirmation that joint treatment of seeds led to increased drought tolerance was a strength.
The primary weakness of the experimental design is the omission of Si-only and DCY84T-only treatments—this makes it difficult to put the effects of the combined treatment into context. Secondarily the decision to look at gene expression only at three weeks seems questionable given that the phenotypic effects are weaker at three weeks than they are earlier. I suppose these weakness could be mitigated somewhat by 1) a few sentences justifying the choice of treatments and sampling times and 2) inclusion of additional details about the prior studies (references 26 and 27) mentioned in lines 62-68 and elsewhere.
The “Results” section has quite a bit of material that ordinarily goes into a “Discussion” section and the “Discussion” is very short and the material in this section is probably not vital to the manuscript. The “Conclusions” section contains material that would normally go into a Discussion, in my opinion. Overall, however, the mixture of elements of a traditional Discussion in the Results did not distract from the readability of the manuscript.
Table 1 would be more useful if results of means comparisons were shown (t show which differences were significant). Figure 3 is not very informative because it is difficult to distinguish sizes of circles. I have never heard of a “hyper p-value” but this could be my ignorance. Supplementary Figure S1 lacks a legend, so I am not sure what it represents although it is referenced in the text. Table 2 contains some undefined acronyms and does not stand on its own.
In the Materials and Methods, I think more details about the treatment of seeds is needed (in particular, what was the “binder solution” used?). The explanation of the drought stress treatment is not sufficiently detailed.